# Integrated Biological Control Strategies for Citrus Rust Mites: Distribution, Impact on Mandarin Quality, and the Efficacy of *Amblyseius largoensis*

**DOI:** 10.3390/insects15110837

**Published:** 2024-10-25

**Authors:** Syed Usman Mahmood, Xiaoyi Huang, Runqian Mao, Huihua Hao, Xiaoduan Fang

**Affiliations:** 1Guangdong Key Laboratory of Animal Conservation and Resource Utilization, Guangdong Public Laboratory of Wild Animal Conservation and Utilization, Institute of Zoology, Guangdong Academy of Sciences, Guangzhou 510260, China; syedusmanmahmood3@gmail.com (S.U.M.); maorun@giz.gd.cn (R.M.); 2Department of Biological Sciences, School of Science, Qiongtai Normal University, Haikou 571127, Chinahhhwhb9901@163.com (H.H.)

**Keywords:** *Amblyseius largoensis*, biochemical analysis, citrus rust mite, biological control, integrated pest management, mandarin orchard

## Abstract

Citrus rust mites are small pests that cause damage to citrus fruits, affecting their quality and the overall harvest. This study focused on understanding where these mites are most commonly found in mandarin orchards and how they impact the fruit’s nutrients, such as vitamin C and sugar levels. This research also tested a natural way to control the mites by using a helpful predatory mite called *Amblyseius largoensis* (Muma). This study took place in a mandarin orchard in Guangzhou, China. It was found that the southern and western parts of the tree had the highest numbers of rust mites, and infested fruits had lower vitamin C levels and higher acidity. *A. largoensis* was effective at controlling the rust mites, especially where mite numbers were moderate. These findings are important because they help farmers target specific areas of the orchard for pest control and show that biological methods like using predatory mites can reduce the need for chemical pesticides. This contributes to healthier fruit production and more sustainable farming practices.

## 1. Introduction

Citrus rust mites (CRMs), *Phyllocoptruta oleivora* Ashmead, particularly prevalent in citrus-growing regions worldwide, are among the most damaging pests in citrus orchards, posing a significant threat to the citrus industry [1]. These mites feed on the epidermal cells of citrus fruits and leaves, resulting in a condition known as russeting or bronzing, which not only diminishes the fruit’s visual appeal but also adversely affects its internal quality [2,3]. The economic implications of CRM infestations are profound, as they directly impact both yield and marketability. Studies have shown that the external damage caused by these mites can lead to substantial reductions in the commercial value of the fruit, with severe infestations causing yield losses of up to 30% in some regions [4,5].

The significance of maintaining high-quality mandarin fruits extends beyond economic considerations, as these fruits are also valued for their nutritional content, particularly vitamin C, which is crucial for consumer health [6]. The deterioration of fruit quality due to pest infestations, therefore, has broader implications for food security and public health. The use of chemical pesticides has traditionally been the primary method for managing CRMs, but this approach is increasingly criticized for its negative environmental impact, including the development of pesticide resistance, harm to non-target organisms, and contamination of soil and water resources [7]. The search for sustainable pest management strategies has thus become a critical area of research, with biological control agents like *Amblyseius largoensis* Muma emerging as promising alternatives [8].

*Amblyseius* spp. are predatory mites that have demonstrated potential in controlling populations of citrus pest mites, particularly in integrated pest management (IPM) programs [9]. Unlike chemical pesticides, biological control agents such as *A. largoensis* offer the advantage of reducing pest populations in an environmentally friendly manner, without the risk of resistance development or non-target species impact [10]. Research has shown that *A. largoensis* is effective in preying on a range of mite species, making it a versatile agent in the biological control arsenal [11,12]. However, evaluation of *A. largoensis* in field conditions, particularly in relation to its functional and numerical response to varying CRM densities, remains underexplored. This gap in knowledge underscores the need for comprehensive studies that assess the ecological interactions between *A. largoensis* and CRMs in real-world orchard environments.

The problem of CRM infestation is further compounded by the complex ecological dynamics within citrus orchards where multiple factors, including environmental conditions and the presence of other pest species, influence the effectiveness of biological control measures [5,13]. Understanding the distribution patterns of both CRMs and natural predatory mites is essential for optimizing control strategies. For instance, the spatial distribution of mites within an orchard can vary significantly, with certain microhabitats supporting higher mite populations [14]. Targeting these hotspots with biological control agents could enhance the effectiveness of pest management programs, but this requires detailed ecological data that are often lacking in the current literature [15,16].

The biochemical impact of CRM infestations on mandarin fruit quality also remains a critical concern. Previous studies have documented significant changes in the biochemical composition of citrus fruits subjected to mite damage, including reductions in vitamin C content and increases in acidity, which compromise both the nutritional value and taste of the fruit [1,4]. These changes can have far-reaching implications for both consumer health and the economic viability of citrus production, particularly in regions where high-quality mandarin fruits are a key agricultural product. Despite these concerns, there is limited research on how varying levels of CRM infestation affect the biochemical properties of mandarin fruits, highlighting another area where further investigation is needed [17].

Given these challenges, this research is driven by the need to develop more effective and sustainable pest management strategies for citrus orchards, particularly those focused on biological control. By addressing the ecological dynamics of CRM populations, the impact of infestations on mandarin quality, and the efficacy of *A. largoensis* as a biological control agent, this study aims to fill critical gaps in the current understanding of citrus pest management. The insights gained from this research could contribute to the development of integrated pest management programs that are both environmentally sustainable and economically viable.

The objectives of this study are threefold: first, to map the distribution patterns of CRMs and natural predatory mites in mandarin orchards; second, to assess the biochemical changes in mandarin fruits subjected to varying levels of CRM infestation; and third, to evaluate the predatory efficiency and reproductive response of mass-rearing the predatory mite *A. largoensis* under different prey density conditions. These objectives are designed to provide a comprehensive understanding of the interactions between CRMs, their natural predators, and the quality of mandarin fruits, with the ultimate goal of enhancing the effectiveness of biological control strategies in citrus orchards.

## 2. Materials and Methods

### 2.1. Study Area and Survey Design

#### 2.1.1. Description

This study was conducted in a mandarin orchard (*Citrus reticulata* Blanco) located in the Zengcheng District of Guangzhou City, Guangdong Province, China (23.346816° N, 113.672909° E). This region is characterized by a subtropical monsoon climate, which provides favorable conditions for citrus cultivation, including warm temperatures and high humidity levels that are conducive to the proliferation of CRMs (*P. oleivora*) and other pest species. The selected orchards spanned an area of approximately 10 hectares and were representative of typical citrus farming practices in the region, including the use of integrated pest management (IPM) strategies. The choice of this location was strategic, given the known prevalence of CRMs in the area and the potential for implementing biological control measures using predatory mites.

The survey design was structured to provide a comprehensive assessment of the spatial distribution of CRMs and their natural enemy predatory mites across the orchards. The orchards were divided into multiple sections based on cardinal directions (north, south, east, and west) and tree height levels (upper, middle, and lower canopy). This stratified sampling approach allowed for the systematic collection of data across different microhabitats, ensuring that variations in environmental conditions, such as sunlight exposure and humidity, were accounted for in the analysis. Each plot comprised a fixed number of trees, randomly selected to minimize sampling bias. The distribution of CRMs was then mapped within these plots to identify potential hotspots of infestation.

The survey was conducted on 24 November 2023 and 1 February 2024 in the selected 3 orchards of Zengcheng District covering both peak and low mite activity periods in Guangzhou. This survey was crucial for capturing spatio-temporal variations in mite populations, which are known to fluctuate based on climatic factors and the phenological stages of the mandarin trees. The data collected provided insights into the ecological dynamics within the orchards, informing the subsequent evaluation of biological control strategies.

#### 2.1.2. Data Collection

Data on CRM populations and environmental conditions were collected using a combination of direct sampling and observational techniques. For mite population assessments, leaves were collected from the selected 15 trees. We used a zigzag method to select 15 trees, and from each tree, 11 leaves were collected: four leaves from the outer and inner canopy of each quadrant (east, west, north, and south) and three from the upper, middle, and lower canopy levels. The leaves were then placed in sealed plastic bags and brought to the lab for counting CRMs and predatory mites under a microscope. The number of CRMs per leaf was recorded, providing a quantitative measure of infestation levels.

Additionally, data on environmental conditions, including temperature and relative humidity, were recorded using a digital Temperature & Humidity meter (Model: COS-03-SJLY) placed at respective heights of selected leaves within the tree canopy. These variables were monitored throughout the survey period to determine their influences on mite population dynamics. The collected data were subjected to statistical analysis to identify significant correlations between mite populations and environmental factors. Mite population densities were assessed in relation to directional orientations (east, west, north, south) and canopy levels (top, middle, bottom) of the citrus trees. This analysis helped identify specific areas of the trees and orchard that were more prone to mite infestations, revealing hotspots based on directional and canopy positioning guiding the development of targeted biological control strategies.

### 2.2. Biochemical Analysis of CRM-Infested Mandarins

#### 2.2.1. Description

To evaluate the biochemical impact of CRM infestations on mandarin fruits (*Citrus reticulata* Blanco), a systematic process of sample collection and analysis was employed. Mandarin samples were harvested from the Zengcheng District orchard, where the survey on mite populations was conducted. The sampling focused on selecting fruits that exhibited varying levels of CRM infestation, ranging from no visible damage (control) to severe russeting (Figure 1). Fruits were categorized based on a visual rating scale adapted from the CODEX quality standards [18] which assess the severity of surface damage caused by CRMs feeding. This rating scale allowed for the classification of the samples into four distinct groups: control (no visible damage), mild infestation (11–50% surface russeting), moderate infestation (51–90% surface russeting), and severe infestation (more than 90% surface russeting). This stratified sampling approach ensured a comprehensive representation of the infestation spectrum within the orchard.

Following the visual assessment, the selected mandarin fruits were transported to the laboratory for biochemical analysis. The focus of the analysis was on key nutritional parameters that are known to be affected by CRM infestations, including vitamin C content, acidity (measured as total titratable acids), soluble solids content (SSC), and sugar content. The samples were processed immediately upon arrival at the laboratory to prevent any post-harvest alterations in their biochemical composition. Samples were analyzed, and the results were recorded to assess the correlation between CRM infestation levels and changes in these biochemical properties.

#### 2.2.2. Laboratory Procedures

The biochemical analyses were conducted using standardized laboratory techniques to ensure the accuracy and reproducibility of the results. Vitamin C content was measured based on GB 5009.86-2016 available online www.GB-GBT.cn (accessed on 11 January 2024), a national food safety standard determination of ascorbic acid in foods using the 2,6-dichlorophenolindophenol (DCPIP) titration method, which is a widely accepted procedure for quantifying ascorbic acid in fruit samples (AOAC, 2012). In this method, the fruit juice extracted from each sample was titrated with DCPIP solution until a stable pink color persisted, indicating the endpoint of the reaction. The amount of DCPIP used was then correlated with the ascorbic acid content in the sample, expressed in milligrams per 100 g of fresh weight. This method was chosen for its precision and sensitivity, particularly in detecting variations in vitamin C levels in response to pest-induced oxidative stress [18].

Acidity, based on GB 12456-2021, the People’s Republic of China’s national food safety standard, was determined by titrating the fruit juice against a standard sodium hydroxide solution (0.1 N) using phenolphthalein as an indicator. The titratable acidity was calculated as grams of citric acid per liter of juice, which is the predominant organic acid in mandarins. This parameter provides insights into the sourness of the fruit, which can be affected by CRMs feeding due to the disruption of the fruit’s metabolic processes.

Soluble solids content (SSC) was measured based on the People’s Republic of China’s agricultural industry standard NY/T 2637-2014, a determination of soluble solids content of fruits and vegetables using the refractometer method.

Soluble sugar was measured based on GB 5009.8-2023, a national food safety standard determination of fructose, glucose, sucrose, maltose, and lactose in food. 

The determination of trace elements was based on the national standard of the People’s Republic of China’s GB 5009.268-2016, a national food safety standard determination of multiple elements in food.

The refractometer provides a direct reading of SSC as a percentage, which is crucial for determining the sweetness of the fruit. Higher CRM infestations were hypothesized to correlate with lower SSC values due to the mites’ impact on the fruit’s ability to synthesize and store carbohydrates.

All biochemical measurements were performed in triplicate to ensure the reliability of the data. The results were statistically analyzed to identify significant differences between the infestation levels, using analysis of variance (ANOVA) followed by post hoc comparisons with Tukey’s HSD test (SPSS Inc., Chicago, IL, USA).

### 2.3. Experimental Evaluation of A. largoensis

#### Description

The experimental evaluation of *A. largoensis* was conducted to determine its predatory efficacy against CRMs (*P. oleivora*) under controlled laboratory conditions. The setup aimed to replicate conditions similar to those found in citrus orchards, allowing for precise control over prey densities and the monitoring of predatory behaviors. The experiments were performed in a climate-controlled chamber set to 25 ± 2 °C with 75 ± 5% relative humidity and a L:D = 16:8 h light–dark cycle, which are optimal conditions for both predator and prey species [19].

Experimental arenas were prepared using Petri dishes (9 cm in diameter) lined with a moist cotton pad to maintain humidity levels and prevent the mites from escaping. A small piece of single citrus leaf disk was placed at the center of each arena, serving as the substrate for the mites (Figure 2). Adult female *A. largoensis* were obtained from laboratory-reared colonies at the Institute of Zoology, Guangdong Academy of Sciences, Guangzhou, maintained on bran-infested flour mites, *Tyrophagus putrescentiae* (schrank). Each experimental arena was stocked with a single adult female *A. largoensis* following a 24 h starvation period to standardize hunger levels across all individuals.

To evaluate the functional response of *A. largoensis*, five different prey densities were used as follows: 2, 4, 8, 16, and 32 individuals of *P. oleivora* per arena. Each prey density was replicated ten times to ensure sufficient statistical power. After 24 h of exposure, the number of consumed prey was recorded using a stereo microscope (Nikon SMZ1000, Nikon, Tokyo, Japan). Each arena was supplemented with a number of consumed CRMs every day. Additionally, the number of eggs laid by the female *A. largoensis* during the experiment was removed every day and recorded to assess the predator’s numerical response. These dual observations provided a comprehensive assessment of the predator’s potential for controlling CRM populations through both predation and reproduction.

### 2.4. Statistical Analysis

The data obtained from the experiments were analyzed using both functional and numerical response models to quantify the predatory efficiency of *A. largoensis*. The functional response, which characterizes the relationship between prey density and the rate at which prey are consumed, was modeled using the Holling Type II functional response equation:(1)$$Ne=\frac{a\cdot N}{1+a\cdot Th\cdot N}$$
where

*Ne* is the number of prey consumed;*a* is the attack rate (i.e., the rate at which the predator encounters and attacks prey);*N* is the prey density;*Th* is the handling time (i.e., the time spent capturing, killing, and consuming a single prey item) [20].

This model was chosen because it typically describes the scenario in which the rate of prey consumption increases with prey density but eventually plateaus due to limitations in handling time. The parameters *a* and *Th* were estimated using non-linear regression analysis, which allowed for precise characterization of the predator’s efficiency at varying prey densities [21].

For the numerical response, which examines how the predator’s reproductive output changes with prey density, a hyperbolic regression model was applied:$$\mathrm{Egg}\;\mathrm{Production}\;(y)=\frac{a\cdot x}{b+x}$$
where

*y* is the number of eggs laid by each female *A. largoensis*;*x* is the prey density;*a* represents the maximum potential egg production rate;*b* is the prey density at which half the maximum egg production rate is achieved.

The relationship between prey density and egg production was further analyzed using Pearson’s correlation coefficient to determine the strength of the correlation. All statistical analyses were conducted using SPSS software (SPSS Inc., Chicago, IL, USA), ensuring rigorous and reliable results.

## 3. Results

### 3.1. Ecological Niche and Distribution Pattern

The survey conducted across the mandarin orchard in Zengcheng District revealed significant variability in the distribution of CRMs and their natural enemy predatory mites across different positions of the orchard. Table 1 summarizes the mean population densities (±SD) of CRMs and predatory mites (PMs) across various sections of a citrus orchard during two surveys, conducted on 24 November and 01 February. It provides an overview of CRM populations on both inner and outer leaves from the four cardinal directions (east, north, west, south) as well as different canopy levels (top, mid, bottom), along with corresponding PM populations.

The CRM populations varied significantly between the two surveys. On 24 November, the highest CRM density was observed on the mid canopy leaves (16.33 ± 56.15), followed by the north outer leaves (11.8 ± 21.07) and the west outer leaves (7.6 ± 23.18). In contrast, the south outer leaves (0.13 ± 0.35) and inner leaves (0.6 ± 1.12) had the lowest CRM populations. Predatory mites (PMs) were almost absent during this survey, except for a low population of 0.13 ± 0.52 on the north inner and south inner leaves.

By 01 February, the CRM populations had increased in several areas, with the highest densities recorded on the north inner leaves (44.47 ± 59.84), west outer leaves (36.6 ± 92.10), and east outer leaves (18.53 ± 39.55). The lowest populations were observed on the south outer (1.4 ± 1.80) and south inner leaves (1.87 ± 2.61). In comparison, PM populations remained consistently low, with no significant changes observed between the two surveys. Specifically, most sections recorded zero PM populations on 24 November. However, by 1 February, slight increases were noted in several areas, including 0.07 ± 0.26 predatory mites in the north outer leaf, mid canopy, and bottom canopy leaves, as well as 0.13 ± 0.52 in the west inner leaf, as detailed in Table 1. 

The results regarding temperature and relative humidity given in Table 2 showed a significant increase in CRM populations from November to February, correlating with changes in temperature and relative humidity. In November, temperatures ranged from 26.84 °C to 27.42 °C, while CRM populations were generally low across all sections. For instance, in the east outer leaves, the CRM population was 0.87 ± 2.10, and in the west outer leaves, it was 7.6 ± 23.18. However, by February, as temperatures dropped to 18.96 °C and 23.72 °C, the CRM populations increased significantly, particularly in areas like the east outer leaves, where the population surged to 18.53 ± 39.55, and the west outer leaves, where the CRM population rose to 36.6 ± 92.10. This suggests that cooler temperatures may favor CRM proliferation.

In addition to temperature changes, relative humidity increased across all sections between the two surveys, from 50.49% to 53.57% in November and from 58.27% to 68.95% in February. This rise in humidity was accompanied by a corresponding increase in CRM populations. For example, in the north inner leaves, relative humidity increased from 52.33% to 64.95%, while the CRM population dramatically rose from 1.07 ± 2.46 in November to 44.47 ± 59.84 in February. Similarly, in the west outer leaves, the humidity rose from 52.24% to 65.98%, coinciding with a sharp increase in CRM population from 7.6 ± 23.18 to 36.6 ± 92.10.

Overall, the findings indicate that both lower temperatures and higher relative humidity during the winter months likely contributed to the substantial increase in CRM populations across various sections of the orchard.

These data indicate an overall increase in CRM population density across most citrus tree positions between the two surveys, while predatory mite populations remained minimal throughout the study period. The data indicate that as the CRM population increased in the second survey, a slight increase in predatory mite populations was also observed, particularly in the north outer, mid canopy, and bottom canopy leaves. The rising CRM densities, especially in the northern and western positions of the citrus trees, suggest potential hotspots requiring targeted pest management interventions.

### 3.2. Changes in Ingredients of CRM-Infested Mandarins

The biochemical analysis of mandarins subjected to varying levels of CRM infestation revealed distinct alterations in the fruit’s nutritional composition, particularly in terms of vitamin C content, soluble solids, soluble sugar, total acids, and mineral content (Table 3). All ingredients measured were affected by CRM infestation; however, only calcium levels exhibited a significant change, indicating that while there were impacts on overall nutritional quality, calcium was particularly influenced by the pest.

#### 3.2.1. Vitamin C Content

Vitamin C levels showed a slight but noticeable decrease as the severity of CRM infestation increased. In the control group (Group A), which had no visible infestation, the mean vitamin C content was recorded at 19.27 mg/100 g. In contrast, mandarins in Group B (11–50% infestation) exhibited a slight decrease in vitamin C content to 18.37 mg/100 g. As the infestation severity increased to above 50% (Group C), the vitamin C content showed a marginal increase to 19.30 mg/100 g, which might suggest a complex biochemical response to moderate levels of stress. However, in the most severely infested mandarins (Group D, above 90% infestation), the vitamin C content dropped again to 18.47 mg/100 g. These variations, although not statistically significant (*p* = 0.8673), suggest a trend where severe infestation levels generally lead to a reduction in vitamin C, which could negatively impact the nutritional quality of the fruit.

#### 3.2.2. Soluble Solids and Sugar Content

The soluble solids content, which is closely associated with the sweetness and taste of the fruit, also varied with CRM infestation levels. The control group (A) had a mean soluble solids content of 11.07%, which slightly decreased in Group B (10.50%) and remained almost constant in Group C (10.73%) and Group D (10.67%). Similarly, the soluble sugar content showed a decrease from 9.03% in the control group to 8.00% in Group B, with a slight increase to 8.90% in Group C and then a minor decline to 8.77% in Group D. Although these changes were not statistically significant, they indicate a trend where CRM infestation, particularly at moderate to high levels, can slightly reduce the sweetness and overall flavor profile of mandarins.

#### 3.2.3. Total Acidity

Total acidity, an important factor influencing the tartness of the fruit, increased with higher levels of CRM infestation. The control group had a mean total acidity of 6.08 g/kg, which rose to 7.66 g/kg in Group B (11–50% infestation). The total acidity slightly decreased to 6.97 g/kg in Group C (above 50% infestation) but increased again to 7.16 g/kg in Group D (above 90% infestation). 

#### 3.2.4. Mineral Content (Zinc, Calcium, Manganese)

The mineral content of the mandarins also varied with the severity of CRM infestation. Zinc levels showed a minor increase from 51.57 mg/kg in the control group to 59.73 mg/kg in Group C, with a slight decrease in Group D to 56.70 mg/kg. Calcium levels demonstrated a significant increase, rising from 265.33 mg/kg in the control group to 301.00 mg/kg in the most severely infested Group D. Manganese content remained relatively stable across all groups, with minor fluctuations.

### 3.3. Efficacy of A. largoensis in Controlling CRMs

The efficacy of *A. largoensis* in controlling CRMs (*P. oleivora*) was evaluated through both functional and numerical response models, which provided a comprehensive understanding of the predatory behavior and reproductive output of *A. largoensis* at varying CRM densities.

#### 3.3.1. Predation Rates

The functional response of *A. largoensis* to different densities of CRMs was characterized by a Type II functional response, where the rate of predation increased with prey density but eventually plateaued as illustrated in Figure 3. The regression analysis supports this, with the coefficients presented in Table 4: the linear (b = −0.474, *p* = 0.0187), quadratic (c = 0.32, *p* = 0.0146), and cubic (d = −0.001, *p* = 0.0224) terms illustrate how predation rates change as prey density increases. Initially, the predation rate increases with density, but it plateaus due to the predator’s handling time constraints at higher densities. The intercept (1.230) reflects baseline predation levels, while the negative linear coefficient suggests that *A. largoensis*’s efficiency decreases slightly at very high prey densities.

The attack coefficient (b) was estimated at 0.749 ± 0.380, with a 95% confidence interval (CI) of −0.461 to 1.960, while the handling time (Th) was 0.170 ± 0.048, with a 95% CI of 0.18 to 0.321. These estimates, derived from the functional response model (R^2^ = 0.795), indicate that *A. largoensis* is capable of efficiently attacking prey at moderate densities, but as prey density increases, handling time becomes a limiting factor in predation efficiency.

These findings demonstrate the effectiveness of *A. largoensis* as a biological control agent, particularly at moderate prey densities, where it can significantly reduce CRM populations. This insight into predator–prey dynamics reinforces the potential of *A. largoensis* in integrated pest management strategies for citrus orchards.

At the lowest prey density of 2 CRMs per arena, *A. largoensis* consumed an average of 1.1 ± 0.18 mites per day. As the prey density increased to 4 CRMs per arena, the predation rate rose to 2.5 ± 0.27 mites per day. At a density of 8 CRMs per arena, the predation rate further increased to 3.7 ± 0.30 mites per day. The highest predation rate was observed at 16 CRMs per arena, where *A. largoensis* consumed an average of 5.2 ± 0.33 mites per day. However, when the prey density was increased to 32 CRMs per arena, the consumption rate slightly decreased to 4.8 ± 0.35 mites per day, suggesting that the predator’s handling time limited its ability to consume more prey as the density increased.

These findings indicate that *A. largoensis* is an effective predator of CRMs, particularly at moderate prey densities, where it can significantly reduce mite populations. The predation rates at different densities highlight the potential of *A. largoensis* as a biological control agent in citrus orchards, especially in scenarios where CRM populations are within the range that maximizes predation efficiency.

#### 3.3.2. Reproductive Output

The numerical response of *A. largoensis* was also evaluated by measuring the number of eggs laid per day by female mites at different CRM densities. A strong positive correlation was observed between prey density and reproductive output, with a correlation coefficient of R^2^ = 0.835 as illustrated in Figure 4. At the lowest prey density of 2 CRMs per arena, *A. largoensis* females laid an average of 0.07 ± 0.02 eggs per day. As the prey density increased to 4 CRMs per arena, the egg production rate increased to 0.12 ± 0.03 eggs per day. The highest egg production rate was observed at 16 CRMs per arena, with an average of 0.14 ± 0.03 eggs per day. Beyond this prey density, no significant increase in egg production was observed, indicating a plateau in reproductive output.

These results suggest that *A. largoensis* not only increases its predatory activity with rising prey densities but also enhances its reproductive output, making it a potential effective biological control agent in environments where CRMs are moderately abundant.

## 4. Discussion

This study was conducted to explore the ecological interactions between CRMs and their natural predator the predatory mite, assess the impact of CRM infestations on the biochemical composition of mandarin fruits, and evaluate the efficacy of *A. largoensis* as a biological control agent under varying prey densities. The results provided valuable insights into the distribution patterns of CRMs and predatory mites, the biochemical changes in infested mandarins, and the functional and numerical responses of *A. largoensis*. In this discussion, the findings are interpreted, compared with the existing literature, their implications are explored, and future recommendations are provided.

Our findings challenge previous research that links CRM proliferation to lower humidity [22,23,24]. In our study, CRM populations increased in areas with higher relative humidity, such as the north inner leaves, where the population rose from 1.07 ± 2.46 to 44.47 ± 59.84 as humidity increased from 52.33% to 64.95%. This suggests that elevated humidity may favor CRM growth, contradicting the traditional view and emphasizing the need to reconsider environmental management strategies in citrus orchards. Based on our findings, while many reports suggest that CRMs thrive in warm, dry conditions, our results did not show a significant correlation between temperature or humidity and CRM population density. We infer that sunlight exposure may be a more influential factor in the proliferation of CRMs in our study. However, our study also introduces new insights into CRM distribution. We observed that CRM populations were consistently higher in the bottom canopy compared with the top canopy, highlighting the impact of microhabitat variability. For instance, CRM densities in the bottom canopy were 1.33 ± 4.37 on 24 November and 8.67 ± 16.12 by 1 February, whereas the top canopy had lower densities (2.53 ± 9.81 and 2 ± 4.52, respectively). These findings underscore the importance of considering both environmental and microhabitat factors for effective pest management. These lower densities might be attributed to less favorable environmental conditions, such as reduced sunlight and which may limit mite reproduction and survival. The bottom canopy recorded the highest CRM densities, while the top canopy consistently had the lowest densities, highlighting the influence of microhabitats on mite distribution. The lack of a strong presence of predatory mites in areas with high CRM densities further suggests that targeted releases or augmentation may be required to enhance its effectiveness as a biological control agent in these orchards. Also, the sparse distribution of predatory mites suggests that natural predation may be insufficient to control CRM populations effectively under current conditions, indicating the need for screening potential predatory mites and targeted releases or augmentation strategies to enhance biocontrol efficacy in controlling CRM populations.

The biochemical analysis revealed that vitamin C content in mandarins decreased as CRM infestation severity increased. Although these variations were not statistically significant, the trend suggests that severe infestation levels generally lead to reduced vitamin C content, potentially impacting the fruit’s nutritional quality. Vitamin C content was of particular interest due to its importance in human nutrition and its sensitivity to oxidative stress caused by pest damage [25]. Research on other phytophagous mites has also indicated that predatory mites, such as *A. largoensis*, play a significant role in managing pest populations, potentially mitigating the impact of oxidative stress on crops [26,27]. Previous studies have reported similar reductions in vitamin C content in fruits subjected to pest damage due to oxidative stress and metabolic disruption [28,29,30,31]. The slight increase in vitamin C content observed at moderate infestation levels could reflect a temporary stress response, boosting the fruit’s antioxidant defenses, but this effect diminishes as the infestation becomes more severe.

The soluble solids and sugar content, closely associated with fruit sweetness, also decreased with increasing CRM infestation levels. These changes, though not statistically significant, indicate that CRM infestation, particularly at moderate to high levels, can reduce the sweetness and overall flavor profile of mandarins. This is consistent with findings from other studies on pest-infested fruits, where pest damage interfered with the fruit’s ability to synthesize and store carbohydrates, leading to lower soluble solids and sugar content [32,33,34].

Total acidity increased with higher levels of CRM infestation, likely resulting in a tarter flavor that may not be desirable to consumers and could affect the fruit’s marketability. Our results showed that the total acids were highest in fruits with 11–50% infestation (7.66%), followed by fruits with more than 90% infestation (7.16%) and 51–90% infestation (6.97%), with the control group having the lowest total acids (6.08%). These findings highlight the impact of CRM infestation on the total acid content in citrus fruits. In comparison, similar trends in acid content have been reported by Rincón-Barón et al. [35], although their study found slightly different acid levels in varying infestation conditions. The observed changes in acidity are consistent with findings that pest feeding disrupts the fruit’s metabolic processes, altering the balance of organic acids in the fruit [35,36].

Mineral content, particularly zinc and calcium, also varied with CRM infestation severity. Zinc levels increased slightly with higher infestation levels, while calcium levels showed a significant increase in the most severely infested group. Our results in which aspects are similar to results of those in [30]. This increase in calcium suggests a stress response in the fruit, possibly as a defensive mechanism against CRM damage. These findings align with research showing that pest infestations can trigger stress responses in plants, altering the accumulation of certain minerals [37].

The spatial variability observed in CRM populations is consistent with previous research. Studies by Pascual-Ruiz et al. [38] and Childers and Rodrigues [39] documented similar patterns of spatial variability in mite populations within citrus orchards, where specific microhabitats supported higher mite populations due to favorable environmental conditions. These studies emphasize the importance of understanding spatial distribution to optimize pest management strategies. The higher CRM densities observed in the southern and western sections of the orchard are not in line with reports by Vermaak et al. [40], who noted that CRMs tend to thrive in areas with lower humidity.

The sparse distribution observed in this study mirrors findings from research on the effectiveness of biological control agents in field conditions. Studies by Fadamiro et al. [41] and McMurtry and Croft [9] highlighted the challenges of using predatory mites like *A. largoensis* in field settings, where their distribution and efficacy can be limited by environmental factors and prey availability. This study’s findings suggest that the natural predation pressure exerted by native predatory mites may not be sufficient to control CRM populations effectively without additional interventions, consistent with observations by Fiaboe et al. [42], who noted that biological control effectiveness often depends on carefully managing predator and prey populations.

The observed reductions in vitamin C content in CRM-infested mandarins align with previous studies. Aguilar-Fenollosa et al. [43] and Singh et al. [44] reported that pest damage leads to reductions in vitamin C due to oxidative stress and the disruption of metabolic processes. While a slight increase in vitamin C content at moderate infestation levels may reflect a temporary stress response, where the fruit boosts antioxidant defenses, this phenomenon has been observed under other stress conditions.

Similarly, the reductions in soluble solids and sugar content in this study correspond with findings from Bacelar et al. [45], who indicated that pest infestations impair the fruit’s ability to synthesize and store carbohydrates, reducing sweetness and flavor. The increase in total acidity found here aligns with research by Wu et al. [46] and Bureš et al. [47], who noted that pest feeding can alter the balance of organic acids in fruit, resulting in higher acidity. Moreover, the increase in calcium levels at higher infestation levels, as reported by Borredá et al. [48] and Gutiérrez-Villamil et al. [49], indicates that pest infestations can trigger plant stress responses, altering mineral accumulation as part of the plant’s defense mechanisms.

These biochemical changes, such as reductions in vitamin C, soluble solids, and sugar content, along with increased acidity and calcium levels, highlight the potential impact of pest infestations on the nutritional quality and marketability of citrus fruits. Severe CRM infestations can significantly alter the taste, nutritional value, and post-harvest quality of mandarins, which could have economic implications for growers as consumers may perceive the fruits to be of lower quality.

The findings of this study also hold important implications for citrus orchard management and integrated pest management (IPM) strategies. Identifying CRM hotspots in the southern and western sections of the orchard provides critical data for targeting pest control interventions. Concentrating efforts on these areas can reduce CRM populations more effectively and minimize the impact on fruit quality. Furthermore, the sparse distribution of *A. largoensis* suggests that natural predation alone may not suffice to control CRM populations. Augmented releases of this biological control agent or other strategies may be necessary to enhance its efficacy in the orchard ecosystem.

Our investigation into the functional and numerical responses of *A. largoensis* when preying on the CRMs provides valuable insights for integrated pest management in citrus orchards. Our findings align with the conceptual framework introduced by Solomon [50], distinguishing between functional and numerical responses.

Solomon [50] introduced a classification for the responses of consumers (predators) to the density of their resources (prey), categorizing them into functional and numerical responses. The functional response type characterizes how the consumption rate of individual consumers changes with resource density, while the numerical response type illustrates how the per capita reproductive rate varies with resource density [50]. Functional and numerical responses serve as essential tools for assessing the efficacy of predatory insects and mites [51,52,53,54,55].

The functional response of a predator concerning prey density typically adheres to one of three mathematical models proposed by Holling [20,56,57]. In the Type I functional response, the number of prey killed increases linearly at a constant rate in relation to prey density. For the Type II response, the number of prey killed rises to a maximum (predator saturation), but the proportion of dead prey declines as prey density increases. The Type III response exhibits a sigmoid curve, where the proportion of prey consumed positively correlates with density within a prey density range. Nevertheless, predator saturation occurs at high prey densities [54]. Predators displaying a Type III response can effectively regulate prey populations [58], while those with a Type II response demonstrate efficiency at low prey densities [59].

The Type II functional response exhibited by *A. largoensis* in our study indicates an efficient predation pattern, where the consumption of CRMs increases with higher densities but reaches a plateau at a certain point. This response type, characterized by a saturation level, is consistent with the models proposed by Holling [20,56,57] and Xiao and Fadamiro [54]. Our observations suggest that *A. largoensis* is particularly effective at moderate to higher CRM densities, offering practical implications for pest control strategies.

The numerical response, as modeled by the hyperbolic equation, provides insights into *A. largoensis*’ population dynamics concerning daily oviposition and CRM density. This understanding contributes to the strategic deployment of *A. largoensis* in pest management programs. Furthermore, egg production is relatively low, and *A. largonesis* may need other prey or pollen to improve their egg lays.

This study revealed significant spatial variability in the distribution of CRMs and across mandarin orchards. The highest CRM densities were consistently found in the northern and western positions of the tree, particularly in the inner leaves. The results from this study clearly show that increasing humidity had a direct influence on the rise in CRM populations. As humidity levels increased between November and February, CRM densities spiked, particularly in areas like the north inner leaves, where relative humidity went up from 52.33% to 64.95% and CRM numbers jumped from 1.07 ± 2.46 to 44.47 ± 59.84. This pattern suggests that higher moisture levels create a more suitable environment for these mites. Previous research, such as the work by Mahmood SU et al. [60], Li et al. [61], and Skoracka et al. [62], supports the idea that mites tend to thrive under humid conditions. In fruit crops, including citrus, the combination of high humidity and temperature often leads to larger mite infestations. Our findings reinforce the importance of managing these environmental factors in pest control strategies. By keeping track of humidity levels in orchards, it is possible to anticipate mite population surges and adjust pest management practices accordingly.

Based on the findings of this study, several recommendations for future research and orchard management can be made. First, further studies are needed to explore the factors influencing the distribution and effectiveness of *A. largoensis* in citrus orchards. This could include research on the environmental conditions that favor the predator’s survival and reproductive success, as well as studies on the optimal release strategies for maximizing its impact on CRM populations. Additionally, research on the biochemical mechanisms underlying the changes observed in CRM-infested mandarins could provide valuable insights into how pest damage affects fruit quality and how these effects can be mitigated. In terms of orchard management, the findings of this study suggest that targeted pest management interventions should be implemented in areas with high CRM densities to reduce the impact on fruit quality. This could include the use of augmented releases of *A. largoensis* or other biological control agents, as well as the integration of other pest management strategies such as cultural practices and the use of resistant varieties. Future research should also explore the potential benefits of combining biological control with other IPM strategies to enhance the overall effectiveness of pest management programs in citrus orchards.

## 5. Conclusions

This study has provided valuable insights into the ecological interactions between CRMs and their natural predator predatory mites, the impact of CRM infestations on the biochemical composition of mandarin fruits, and the effectiveness of *A. largoensis* as a biological control agent. The findings revealed significant spatial variability in CRM populations within the orchard, with CRM hotspots identified primarily in the southern and western positions of trees. CRM populations increased in most sections as temperatures dropped from November to February. This suggests that cooler temperatures during the winter months may promote higher mite activity, especially in areas with moderate canopy cover.

Humidity Effect: Higher relative humidity in February correlated with increased CRM populations. This trend suggests that more humid conditions, alongside cooler temperatures, might create favorable environments for CRM proliferation, particularly in areas like the north and west canopy sections where the mite populations were already higher. In summary, both lower temperatures and higher relative humidity during the second survey in February seem to have favored the growth of CRM populations across various positions in the citrus orchard.

The biochemical analysis showed that CRM infestations led to a reduction in vitamin C content, soluble solids, and sugar content in mandarins, alongside an increase in acidity and calcium levels. This study also demonstrated that *A. largoensis* exhibited a functional and numerical response to varying CRM densities, indicating its potential as an effective biological control agent under optimal conditions. The implications of these findings are significant for citrus orchard management and the development of integrated pest management (IPM) strategies. Targeted pest management interventions in CRM hotspots could effectively reduce mite populations and mitigate their impact on fruit quality. The demonstrated efficacy of *A. largoensis* highlights its potential as a crucial component of IPM strategies, offering an environmentally sustainable alternative to chemical pesticides. Additionally, the observed biochemical changes in infested mandarins underscore the need for vigilant pest management to preserve the nutritional value and marketability of citrus fruits. Our study emphasizes the potential of *A. largoensis* as a biological control agent against *P. oleivora*, offering efficiency in oviposition. These findings contribute to the broader knowledge of predator–prey interactions and can guide practical applications in citrus orchard management. Future research should delve into field applications and assess the long-term impact of *A. largoensis* on CRM populations.

Future research should focus on further exploring the environmental factors that influence the effectiveness of *A. largoensis* and the biochemical mechanisms underlying CRM-induced changes in fruit quality. Practical applications of this study include the implementation of targeted biological control strategies in identified CRM hotspots and the integration of *A. largoensis* with other IPM practices to enhance the overall effectiveness of pest management programs in citrus orchards.

## Figures and Tables

**Figure 1 insects-15-00837-f001:**
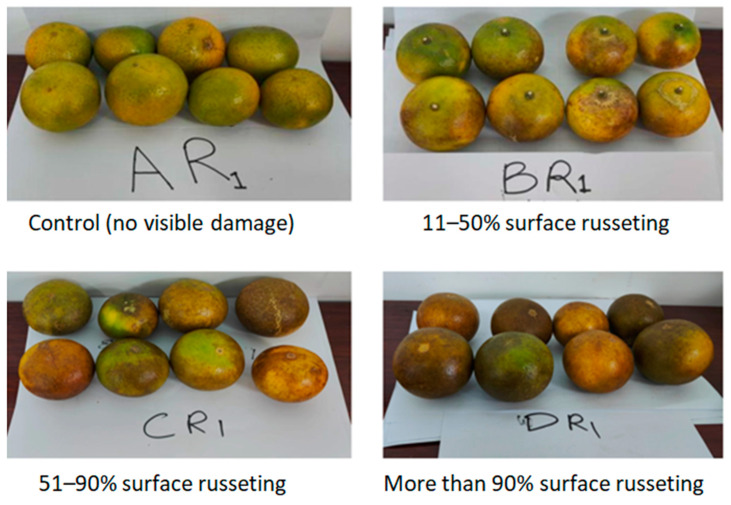
Classification of citrus fruits based on CRM infestation severity using a visual rating scale.

**Figure 2 insects-15-00837-f002:**
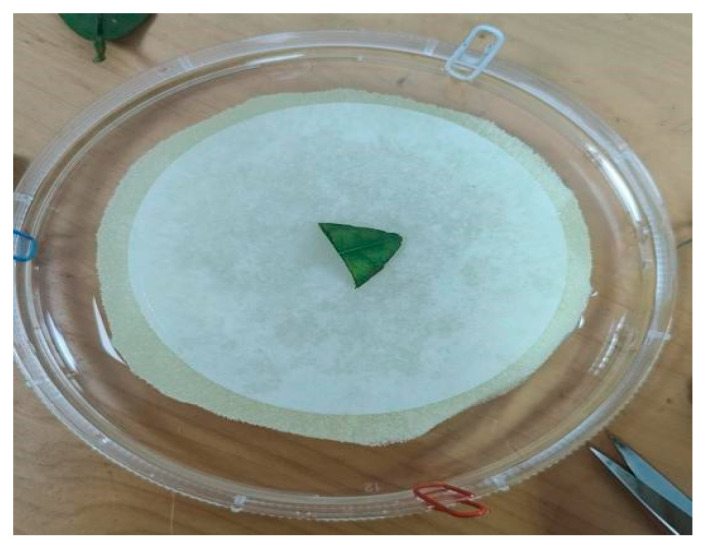
Experimental arena.

**Figure 3 insects-15-00837-f003:**
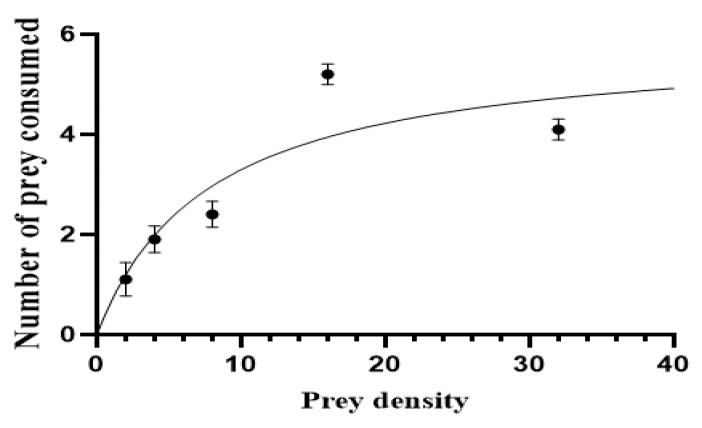
Functional response curve of *A. largoensis* to varying CRM densities. The curve demonstrates the increasing predation rate with rising CRM densities, followed by a plateau as the prey density exceeds 16 CRMs per arena.

**Figure 4 insects-15-00837-f004:**
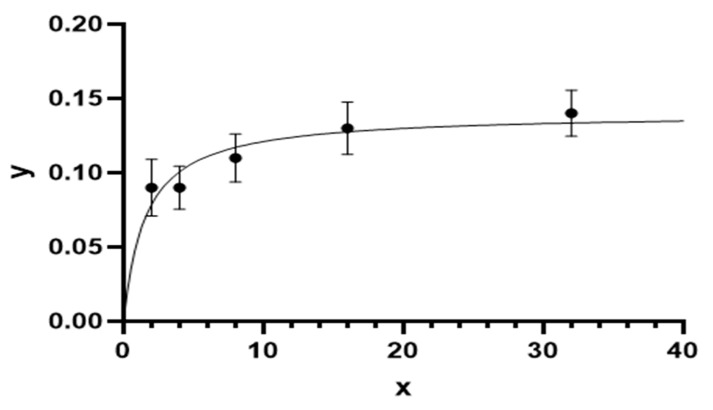
Numerical response of *A. largoensis* adult females to *P. oleivora* on leaf disks, where x = density of *P. oleivora* per arena and y = the number of eggs laid by a *A. largoensis* adult female per day.

**Table 1 insects-15-00837-t001:** Population densities (mean ± SD) of CRMs and predatory mites across different orchard sections based on the data from two surveys.

Citrus Tree Position	CRM Population on 24 November (Mean ± SD)	CRM Population on 01 February (Mean ± SD)	PM Population on 24 November (Mean ± SD)	PM Population on 1 February (Mean ± SD)
East (Outer Leaf)	0.87 ± 2.10	18.53 ± 39.55	0.00 ± 0.00	0.00 ± 0.00
East (Inner Leaf)	0.6 ± 1.12	9.93 ± 18.39	0.00 ± 0.00	0.00 ± 0.00
North (Outer Leaf)	11.8 ± 21.07	17.4 ± 18.70	0.00 ± 0.00	0.07 ± 0.26
North (Inner Leaf)	1.07 ± 2.46	44.47 ± 59.84	0.13 ± 0.52	0.00 ± 0.00
West (Outer Leaf)	7.6 ± 23.18	36.6 ± 92.10	0.00 ± 0.00	0.00 ± 0.00
West (Inner Leaf)	3.8 ± 11.28	12.67 ± 41.12	0.00 ± 0.00	0.13 ± 0.52
South (Outer Leaf)	0.13 ± 0.35	1.4 ± 1.80	0.00 ± 0.00	0.00 ± 0.00
South (Inner Leaf)	0.6 ± 1.12	1.87 ± 2.61	0.13 ± 0.52	0.00 ± 0.00
Top Canopy Leaf	2.53 ± 9.81	2 ± 4.52	0.00 ± 0.00	0.00 ± 0.00
Mid Canopy Leaf	16.33 ± 56.15	4.93 ± 8.73	0.00 ± 0.00	0.07 ± 0.26
Bottom Canopy Leaf	1.33 ± 4.37	8.67 ± 16.12	0.00 ± 0.00	0.07 ± 0.26

PM representative of predatory mite.

**Table 2 insects-15-00837-t002:** Temperature and relative humidity across different orchard sections based on the data from two surveys.

Citrus Tree Position	Temperature (°C) 24 November (Mean ± SD)	Temperature (°C) 1 February (Mean ± SD)	Relative Humidity (%) 24 November (Mean ± SD)	Relative Humidity (%) 1 February (Mean ± SD)
East (Outer Leaf)	27.01 ± 2.87	20.79 ± 0.89	52.17 ± 14.06	61.66 ± 3.28
East (Inner Leaf)	27.07 ± 3.02	21.12 ± 1.06	52.06 ± 13.87	65.65 ± 3.23
North (Outer Leaf)	27.17 ± 3.08	20.07 ± 0.36	52.48 ± 13.72	65.64 ± 2.39
North (Inner Leaf)	27.11 ± 2.97	19.99 ± 0.34	52.33 ±13.95	64.95 ± 1.41
West (Outer Leaf)	26.98 ± 2.87	19.88 ± 0.19	52.24 ± 14.05	65.98 ±1.54
West (Inner Leaf)	26.84 ± 2.68	19.58 ± 0.21	53.57 ± 13.16	64.96 ± 1.78
South (Outer Leaf)	27.03 ± 2.96	18.96 ± 0.30	53.15 ± 13.67	66.47 ± 1.55
South (Inner Leaf)	27.04 ± 2.83	19.02 ± 0.23	53.33 ± 13.31	68.95 ± 3.14
Top Canopy Leaf	27.19 ± 2.96	21.95 ± 1.44	52.97 ± 13.64	62.74 ± 3.73
Mid Canopy Leaf	27.42 ± 2.78	23.72 ± 0.66	53.34 ± 13.98	59.32 ± 3.61
Bottom Canopy Leaf	27.41 ± 2.80	22.59 ± 0.79	50.49 ± 13.31	58.27 ± 1.80

There was no significant difference among the means.

**Table 3 insects-15-00837-t003:** Citrus rust mites damaged mandarin ingredients’ profiles.

Treatment	Vitamin C (mg/100 g)	Soluble Solids (%)	Soluble Sugar (%)	Total Acid (g/kg)	Zn (mg/kg)	Ca (mg/kg)	Mn (mg/kg)
A (Control)	19.27 ± 1.75 a	11.07 ± 0.38 a	9.03 ± 0.32 a	6.08 ± 1.34 a	51.57 ± 1.98 a	265.33 ± 15.98 a	106.7 ± 14.99 a
B (11 to 50%)	18.37 ± 2.53 a	10.5 ± 0.36 a	8 ± 0.1 a	7.66 ± 1.15 a	50.20 ± 5.04 a	174.67 ± 31.51 b	93.57 ± 5.77 a
C (Above 50%)	19.3 ± 1.1 a	10.73 ± 0.35 a	8.9 ± 0.56 a	6.97 ±1.82 a	59.73 ± 12.07 a	278.33 ± 35.13 a	102.3 ± 8.86 a
D (above 90%)	18.467 ± 1.43 a	10.67 ± 0.64 a	8.77 ± 0.61 a	7.16 ± 1.17 a	56.7 ± 7.03 a	301 ± 35.04 a	104.83 ± 5.37 a
P (ANOVA)	0.8673	0.51	0.0822	0.5921	0.4192	0.0044 **	0.41

Means within the same column followed by different letters (a, b) are significantly different at *p* < 0.05. ** Highly significant at *p* < 0.05.

**Table 4 insects-15-00837-t004:** Coefficients of the polynomial linear regression of predation proportion for *A. largoensis* against *P. oleivora* under laboratory conditions.

Parameter	Estimate	SE	t	*p*
Intercept	1.230	0.350	1.985	0.0785
Linear (b)	−0.474	0.92	2.862	0.0187
Quadratic (c)	0.32	0.006	3.015	0.0146
Cubic (d)	−0.001	0.000	2.753	0.0224

Note: The regression equation is as follows: N_e_/*N_o_* = a + b*N_o_* + c*N_o_*^2^ + d*N_o_*^3^ + e.

## Data Availability

The data sheets are available and can be requested from the corresponding author on reasonable request.

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
