# Peer review of "Integrated Biological Control Strategies for Citrus Rust Mites: Distribution, Impact on Mandarin Quality, and the Efficacy of *Amblyseius largoensis"

_insects, 2024, doi:10.3390/insects15110837_

Round 1
Reviewer 1 Report
Comments and Suggestions for Authors
Dear Author,
I have shown my comments on the PDF (attached). My points start on page 7, most of them at the left margin. If you have explanations for that, it is welcome.

It is very good, except a few prepositions that might be missing, which is natural but must be taken care of.
Author Response
Comments 1: [There is no statistical analysis for Table 1.]
Response 1: Thank you for your observation regarding the lack of statistical analysis for Table 1. We appreciate the importance of statistical rigor and would like to clarify that the data in Table 1 are presented as means with standard deviations (S.D.), based on 15 replications for each canopy level and cardinal direction. This information can be found in the revised manuscript on page 7, paragraph 3, lines 284-287. Since each canopy level and direction had 15 replications in every orchard, the mean and S.D. provide a reliable representation of population densities, capturing trends and variability in CRM and predatory mite populations. The primary purpose of Table 1 was to offer a descriptive summary of these population dynamics across various orchard sections, rather than perform formal statistical comparisons. Due to the low and inconsistent predatory mite populations, more complex statistical analyses would not have been suitable in this context. We hope this explanation clarifies the reasoning for presenting the data in this format. We also have supplied some statistical analysis in the manuscript.
Comment 2: [No. Please see Table 1.]
Response 2: Thank you for your comment. We agree with you and after reviewing the data in Table 1, we recognize the need to adjust the phrasing to ensure it accurately reflects the data. The predatory mite (PM) populations remained very low across both surveys, with only slight increases observed in a few specific areas, as shown in Table 1. we will revise the text for clarity and to better align it with the data presented. This change can be found in the revised manuscript on page 7, paragraph 4, lines 293-298.
"In comparison, PM populations remained consistently low, with no significant changes observed between the two surveys. Specifically, most sections recorded zero PM populations on November 24. However, by February 01, slight increases were noted in several areas, including 0.07 ± 0.26 predatory mites in the North Outer Leaf, Mid Canopy, and Bottom Canopy leaves, as well as 0.13 ± 0.52 in the West Inner Leaf, as detailed in Table 1." This revision ensures that the text accurately corresponds to the data in Table 1 and provides a clearer representation of the observed trends. Thank you for your valuable feedback.
Comments 3: [Table 3 is about ingredients, not weather.]
Response 3: Response: Thank you for pointing this out. We agree with this comment. Therefore, We have corrected the reference to the relevant data regarding temperature and relative humidity, which is actually in Table 2. This change can be found in the revised manuscript on page 7, paragraph 5, line 299. Thank you for your valuable feedback, which has helped improve the accuracy of the manuscript.
Comment 4: [Was there no interest to compare the two dates? If that was needed, direction, canopy and date might be taken together as factorial design and see if they had any interaction. The same for Table 2 and 3]
Response 4: Thank you for your insightful comment. While we recognize the potential benefits of a factorial design to compare the two survey dates and assess interactions among direction, canopy, and date, we chose not to pursue this approach in the current study. Our primary focus was on describing the overall trends in CRM and predatory mite populations over time, rather than exploring complex interactions. The low and inconsistent populations of predatory mites across both surveys made it challenging to derive meaningful conclusions from a factorial analysis. Instead, we aimed to provide a clear depiction of the spatio-temporal dynamics observed in the data. However, we appreciate your suggestion and will consider a more detailed factorial design analysis in future research, particularly if the data set allows for it.
Comment 5: [What does Table 2 compare (as shown with letters?) Well, the letter is only "a" and if that was mean comparison, there was no need for it because it was all the same. In that case, it may have to be denoted as an asterisk (footnote) that there was no significant difference. The comparison looks for each colomn. But the columns contain different sources of variation (like direction and canopy. How can these be compared as if they belonged to the same factor? I suggest the removal of the letter and that is it. If ANOVA is needed, it may be done in a factorial ANOVA with two factors - direction and canopy. This can even help capture if there was interaction]
Response 5: Thank you for your insightful comment. We agree that the use of the letter "a" in Table 2 was unnecessary and may have caused confusion. As such, we have removed the letter and added a footnote to indicate that there was no significant difference among the means values presented in the table 2. This change can be found in the revised manuscript on page 8 and 9 in Table 2.
Regarding your suggestion for conducting a factorial ANOVA to analyze the interactions between direction and canopy, we appreciate the recommendation. While this study primarily focused on descriptive trends, we will consider the inclusion of such analyses in future research to better explore these relationships.
Thank you once again for your valuable feedback, which has helped enhance the clarity and accuracy of the manuscript.
Comment 6: [Table 3. Only Ca was significantly affected. So, can we say that ingredients were affected?"]
Response 6: Thank you for your comment. You are correct that while there was an effect on all ingredients, only calcium (Ca) showed a significant change based on the data presented in Table 3. We have revised the text to clarify that all ingredients were affected by CRM infestation, but only calcium levels reached statistical significance.
This change can be found in the revised manuscript on page 9, paragraph 1, lines 337-339:
All ingredients measured were affected by CRM infestation; however, only calcium levels exhibited a significant change, indicating that while there were impacts on overall nutritional quality, calcium was particularly influenced by the pest.
Thank you for your valuable feedback, which has helped enhance the clarity of the manuscript.
Comment 7: [Figure 3 is not clear. Was it correlation or ANOVA? The letters attached to the bars indicate mean separation between treatments. If that was correlation, a table may suffice showing correlation coefficients. If it was not correlation,]
Response 7: Thank you for your comment regarding Figure 3 and we agree with you. To clarify, the data presented in Figure 3 were based on ANOVA results, which are already clearly detailed in Table 3. To avoid repetition and enhance clarity, we have decided to remove Figure 3 from the manuscript.
This adjustment streamlines the presentation of results and ensures that readers can easily reference the statistical findings without confusion. This change can be found in the revised manuscript on page 9. Thank you for your valuable feedback, which has contributed to improving the clarity and organization of the manuscript.
Comment 8: [Table 4 be cited here.]
Response 8: Thank you for your observation. We appreciate your suggestion and agree to include a citation for Table 4 in the relevant text. We have added a reference to Table 4 to ensure clarity regarding the functional response analysis presented. This change can be found in the revised manuscript on page 10, paragraph 6, line 386.
The regression analysis supports this, with the coefficients presented in Table 4:
Thank you again for your valuable feedback, which has helped improve the clarity of the manuscript.
Comment 9: [Do you mean Table 5?]
Response 9: Yes, we agree with you, we meant to reference Table 5 instead of Table 4. We appreciate your attention to detail. The necessary corrections have been made in the revised manuscript to accurately cite Tables throughout the relevant sections. Table 5 has been removed as per suggestion. This change can be found in the revised manuscript on page 11.
Thank you for your valuable feedback!
Comment 10: [Table 5 is unnecessary because all the information is found in the text. Only R-square is not stated and that can be added, of course, in the text to make it complete.]
Response 10: Thank you for your insightful comment. We agree that Table 5 is unnecessary since the information it contains is adequately presented in the text. We have removed Table 5 from the manuscript. Additionally, we have included the R-squared value in the text to provide a complete overview of the regression analysis. This change can be found in the revised manuscript on page 10, paragraph 7, line 394, where we have specified the R-squared value (R² = 0.795).
Thank you again for your valuable feedback, which has contributed to improving the clarity and completeness of the manuscript.
Comment 11: [Should Table 6 and Figures 4 and 5 be cited in the text above? Also, why is the same data shown in Table and Figure at the same time? It is strange. I suggest the Figures be taken.]
Response 11: Thank you for your insightful comments. We agree that Table 6 and Figures 4 and 5 should be cited in the text for clarity. To avoid redundancy in presenting the same data in both table and figure formats, we have decided to remove Table 6 while keeping Figures 4 and 5 in the manuscript which are now named as Figure 3 and Figure 4 on Page 12. Citation of Figure 3 (as illustrated in Figure 3) on page 10, paragraph 6, line 385, while Figure 4 (as illustrated in Figure 4) can be found in the revised manuscript on page 11, paragraph 4, line 421, Thank you for your valuable feedback, which has helped enhance the clarity and organization of the manuscript.
Reviewer 2 Report
Comments and Suggestions for Authors
Dear Editor and authors,
I was between a rock and a hard place when it came to the rating of this article. This article is well-produced, the English is perfect, the presentation is well-done and the literature is adequate. Was the predator Amblyseius largoensis previously released in the orchard? Was it A. largoensis present in the orchard? No mention of an identification was made. The presence in the orchard of it was insignificant. Temperature and humidity changes do not have the same effect on CRM and A. largoensis. How will this be addressed when you release the predator? The results of the laboratory tests of the influence of the pest on the fruit quality show no influence. It was just an indication
The benefit of these results for me was establishing the hot spots of the pest in the orchards, the influence of temperature and humidity on its population and that A. largoensis may be an effective bio-control agent.
Author Response
Comment 1: I was between a rock and a hard place when it came to the rating of this article. This article is well-produced, the English is perfect, the presentation is well-done, and the literature is adequate
Response 1: Thank you for your positive feedback regarding the production quality of the article, the clarity of the English, and the adequacy of the literature. I appreciate your acknowledgment of the effort put into the manuscript.
Comment 2: "Was the predator Amblyseius largoensis previously released in the orchard? Was it A. largoensis present in the orchard? No mention of an identification was made."
Response 2: Thank you for this important comment. We agree with you and in the revised manuscript, we have clarified that the Amblyseius largoensis used in our experiments were sourced from a culture reared in our laboratory at Institute of zoology, Guangdong Academy of Sciences, Guangzhou, as mentioned in the Materials and Methods sub section 2.3.1. Description. While various Amblyseius species are indeed used in IPM (already described in Introduction part page 2, line paragraph 9, line 230), but our focus was specifically on A. largoensis due to its demonstrated efficacy in controlling Phyllocoptruta oleivora Ashmead. We believe that A. largoensis shows significant potential for citrus rust mites (CRM) control, which is supported by the results of our study.
Comment 3: "The presence in the orchard of it was insignificant."
Response 3: Thank you for your observation. While the presence of Amblyseius largoensis in the orchard may have been limited, it is important to note that other species within the same genus are commonly found in these environments. We selected A. largoensis for our study because it has shown potential as an effective predator for controlling Citrus red mites (CRM). Our research aims to evaluate its efficacy and explore the possibility of introducing A. largoensis into orchards as a biological control agent. This focus aligns with our goal of enhancing integrated pest management strategies in citrus production systems.
Comment 4: "Temperature and humidity changes do not have the same effect on CRM and A. largoensis. How will this be addressed when you release the predator?"
Response 4: Thank you for your important comment. In our experiments, the efficacy of Amblyseius largoensis was evaluated under the same temperature and humidity conditions provided to CRM. This approach ensures that the findings are directly applicable to field conditions, where both species will experience similar environmental influences. To address the differing effects of temperature and humidity on these species during predator releases, we plan to monitor environmental conditions closely and choose optimal release times. By leveraging the adaptability of A. largoensis to local climate conditions, we aim to enhance its establishment and effectiveness in controlling CRM populations. This strategy will support integrated pest management practices in orchards, ultimately improving pest control outcomes.
Comment 5: "The results of the laboratory tests of the influence of the pest on fruit quality show no influence. It was just an indication."
Response 5: Thank you for your comment. While the laboratory tests indicated no significant overall influence on fruit quality, it’s important to note that all tested ingredients were affected by CRM infestation. Notably, calcium (Ca) levels showed significant changes as illustrated in Table 3 on Page 9. Additionally, the cosmetic value of the fruit drops significantly due to CRM damage, which can impact marketability even if overall quality appears unaffected. These effects highlight the need for detailed analyses to understand how CRM infestation influences both nutritional and aesthetic aspects of fruit.
We appreciate your feedback, which underscores the importance of recognizing these specific impacts in our assessments.
Comment 6: "The benefit of these results for me was establishing the hot spots of the pest in the orchards, the influence of temperature and humidity on its population, and that A. largoensis may be an effective bio-control agent."
Response 6: Thank you for your valuable feedback regarding the benefits of the study. We have emphasized the identification of pest hotspots and the influence of environmental factors on pest populations. Additionally, we have highlighted the potential of Amblyseius largoensis as an effective biological control agent, reinforcing its relevance in integrated pest management strategies.